# Activation in human auditory cortex in relation to the loudness and unpleasantness of low-frequency and infrasound stimuli

**Oliver Behler** [1] *, **Stefan Uppenkamp** [1,2]

**1** Medizinische Physik, Carl von Ossietzky Universität Oldenburg, Oldenburg, Germany, **2** Cluster of Excellence Hearing4All, Carl von Ossietzky Universität Oldenburg, Oldenburg, Germany

* oliver.behler@uni-oldenburg.de

## Abstract

Low frequency noise (LFS) and infrasound (IS) are controversially discussed as potential causes of annoyance and distress experienced by many people. However, the perception mechanisms for IS in the human auditory system are not completely understood yet. In the present study, sinusoids at 32 Hz (at the lower limit of melodic pitch for tonal stimulation), as well as 8 Hz (IS range) were presented to a group of 20 normal hearing subjects, using monaural stimulation via a loudspeaker sound source coupled to the ear canal by a long silicone rubber tube. Each participant attended two experimental sessions. In the first session, participants performed a categorical loudness scaling procedure as well as an unpleasantness rating task in a sound booth. In the second session, the loudness scaling procedure was repeated while brain activation was measured using functional magnetic resonance imaging (fMRI). Subsequently, activation data were collected for the respective stimuli presented at fixed levels adjusted to the individual loudness judgments. Silent trials were included as a baseline condition. Our results indicate that the brain regions involved in processing LFS and IS are similar to those for sounds in the typical audio frequency range, i.e., mainly primary and secondary auditory cortex (AC). In spite of large variation across listeners with respect to judgments of loudness and unpleasantness, neural correlates of these interindividual differences could not yet be identified. Still, for individual listeners, fMRI activation in the AC was more closely related to individual perception than to the physical stimulus level.

## 1. Introduction

Low frequency sound (LFS: typically applies to frequencies below 200 Hz) and Infrasound (IS: below 20 Hz) emerge from a variety of natural events. However, the abundance of these sounds within our environment has significantly increased with the advance of technical sources such as construction machines, air traffic and industrial wind turbines. Meanwhile, the potential impact of low frequency noise on human health and well-being has become a much debated topic, fueled by many reports of people suffering from annoyance and distress that is attributed to LFS exposure [1].

**Data Availability Statement:** All relevant data are available via the Harvard Dataverse (https://doi.org/10.7910/DVN/IV9VJZ).

**Funding:** OB and SU received funding from the European Metrology Programme for Innovation

and Research (EMPIR - EURAMET), grant number 15HLT03 Ears II, https://www.euramet.org/research-innovation/research-empir/. The funders had no role in study design, data collection and analysis, decision to publish, or preparation of the manuscript.

**Competing interests:** The authors have declared that no competing interests exist.

It has been demonstrated many times that infrasonic tones are audible if the sound level is sufficiently high [e.g. 2–5]. In fact, although the qualitative perception eventually changes from a tonal sensation to a sensation of ″discontinuous, separate puffs″ [6], detection thresholds increase gradually and without sudden shift towards infrasonic frequencies.

Physiological data support the notion that IS and sounds in the typical audio frequency range share similar perceptual mechanisms. For instance, IS-induced changes of distortion product otoacoustic emissions (DPOAE) have confirmed that IS enters the inner ear and may modulate cochlear function [7, 8]. Beyond that, two functional magnetic resonance imaging (fMRI) studies have found increased activation in bilateral auditory cortex (AC) in response to 12 Hz tones (at high sound pressure levels of 110 dB and above), revealing that similarities between IS and ″normal sound″ persist up to early cortical processing [9,10].

Another trend that extends into infrasonic frequencies pertains to the perceived loudness of sounds: With decreasing frequency, loudness continues to grow more steeply as a function of sound pressure level [11–13]. As a result, even small changes of level by only a few decibels above threshold may elicit quite significant changes of the perceived intensity of IS stimuli.

Several studies have also investigated judgments of listeners with respect to annoyance and unpleasantness of LFS and IS under laboratory conditions [overview in 1]. Similar to loudness, the growth of annoyance and unpleasantness with sound level steepens as frequencies decrease [4, 14]. It has however been observed that the close relationship between loudness and annoyance does not hold any more for noise with high levels at frequencies below 100 Hz, where A-weighted levels and loudness estimates underestimate ratings of perceived annoyance [15–17]. In addition, some researchers have reported exceptionally large interindividual variability in the extent of annoyance for low frequency noise [16, 17]. At the very least, there are extreme outliers, as in the case of a group of self-reported ″noise-sufferers″ investigated by Inukai et al. [18]. For this particularly sensitive group of listeners LFS and IS tones deemed unacceptably unpleasant under daily living conditions, even at levels that more or less coincided with their detection thresholds and despite the fact that individual thresholds in this group were comparable to those of a control group.

The questions arising from this and addressed in the present study are: (1) Do perception and neural responses differ between LFS (at the lower limit of pitch perception) and IS tones? (2) Are perceived loudness and unpleasantness distinctly represented in the human brain? (3) Can interindividual differences with respect to loudness and unpleasantness for LFS and IS be identified in terms of objective, physiological correlates?

Previous fMRI studies have demonstrated that, at least for sounds in the typical audio frequency range, activation in AC as measured by means of the blood oxygen level dependent (BOLD) response is more related to individually perceived loudness rather than the physical characteristics of sound alone [review in 19]. A few studies have also investigated the unpleasantness of sounds by means of fMRI. Their results indicate that additional regions not directly associated with the auditory system, such as the amygdala, might be involved in the processing of unpleasantness [e.g. 20]. Other studies suggest that a learned aversive valence for sounds (e.g., through fear conditioning), which might be the cause of a higher unpleasantness in some cases, is reflected by altered AC activity [e.g., 21, 22]. Given that activation in response to LFS and IS appears to be very similar to that of typical audio sound, we hypothesized that fMRI is a suitable tool to disentangle the representation of loudness and unpleasantness and to identify interindividual differences in the perception of LFS and IS.

In the present study, we measured fMRI activation in a group of 20 normal hearing listeners (without high self-reported sensitivity to LFS) in response to an LFS tone at 32 Hz (eliciting a tonal sensation at a very low pitch) and an IS tone at 8 Hz with varying, individually adapted levels. We investigated the measures of activation in relation to estimates of individual

loudness and unpleasantness for the respective stimuli as obtained from psychoacoustic experiments performed in a sound booth as well as in the fMRI scanner. To disentangle the neural representation of sound level, loudness and unpleasantness, we compared different regression models based on cross-validated prediction performance in addition to conventional contrast-based activation maps.

## 2. Methods

### 2.1 Participants

Twenty healthy normal hearing volunteers were recruited at the University of Oldenburg and gave written informed consent to participate in this study. One female participant was excluded from the analysis due to an abnormal condition/signal detected in the structural image. The remaining sample comprised 10 female and 9 male participants, ranging from 21 to 34 years of age (mean: 26 years). All participants had hearing thresholds better than 20 dB HL in the range from 125 to 8k Hz, as tested by means of standard pure tone audiometry with a clinical audiometer and Sennheiser HDA 200 Headphones (Sennheiser electronic GmbH & Co. KG, Wedemark, Germany). A questionnaire was used to ensure that subjects had no conditions contraindicative for MRI. The study was approved by the ethics committee of the University of Oldenburg.

Each participant attended three experimental sessions, with at least one day in between two subsequent sessions. In the first and second session, several psychoacoustical tests were performed in an acoustically shielded sound booth. In the third session, the auditory fMRI experiment was performed.

The psychoacoustic test battery included detection threshold assessments, a categorical loudness scaling procedure, unpleasantness rating tasks as well as a pairwise unpleasantness comparison task. The acoustic stimuli comprised tones at 8, 16, 32, 64, and 128 Hz. Within the scope of this manuscript, we only report loudness judgments and unpleasantness ratings for tones at 8 and 32 Hz, since only these are directly linked to the fMRI experiment. A complete and more detailed report of the behavioral data at all tested frequencies will be provided elsewhere.

### 2.2 Acoustic stimuli

Stimuli were delivered monaurally to the right ear via standard insert foam eartips coupled to a special loudspeaker sound source (described in more detail in [5]) via a long silicone rubber tube. The sound source was connected to a PC setup with external 24-bit DA-converter (UA-25, Edirol by Roland, Hamamatsu, Shizuoka, Japan) and analog power amplifier (BAA 120, TIRA GmbH, Schalkau, Germany). Second-order low-pass filters ($f_c$ = 30 Hz) were inserted between the amplifier and the loudspeaker to keep harmonics sufficiently well below the normal hearing threshold level.

All acoustic stimuli were tone bursts with 1.5 seconds duration, created at a sampling rate of 44.1 kHz and 24 bit depth. Following the recommendations of Kühler et al. [5], $\cos^2$ rise and decay ramps were adapted to each frequency to allow for a minimum number of four periodic oscillations at full amplitude as well as three oscillations for each ramp in order to meet the requirements for narrowband signals. Here, we used ramp durations of 125 ms for the 32-Hz tones and 375 ms for the 8-Hz tones. The maximum presented sound pressure level (upper SPL limit) throughout all experiments was 140 dB SPL.

All experiments were programmed and presented using MATLAB 2014a (Mathworks, Natick, MA) and the Cogent 2000 toolbox (v125, http://www.vislab.ucl.ac.uk/cogent.php, London, UK).

## 2.3 Psychoacoustical measurements

**2.3.1 Adaptive categorical loudness scaling.** Categorical loudness scaling is a psycho-acoustic measurement procedure to assess individual loudness perception [e.g. 23–26]. In this study, an adaptive version of this procedure as proposed by Brand and Hohmann [26] was used. A response scale with 11 response alternatives was presented on a computer screen in front of the participant. The response scale included seven named loudness categories—"inaudible", "very soft", "soft", "medium", "loud", "very loud" and "extremely loud"—and four unnamed intermediate response alternatives. Acoustic stimuli were presented with varying intensity. Following each stimulus, listeners were asked to make a loudness judgment by choosing a response alternative using button presses. The procedure implicitly consisted of three phases. The first phase was designed to roughly estimate the audible dynamic range for each participant. Stimulus intensities were varied systematically until the responses "inaudible" and "extremely loud" were given or until 0 dB SPL or the upper SPL limit were reached. In the second and third phases, more data were collected for sound levels presented within this dynamic range. Taken together, the number of trials varied from 18 to 31 per run, depending on the perceptual dynamic range of the participant for the stimulus, with 23 presentations on average for the 8 Hz tone and 30 presentations for the 32 Hz tone.

In the first experimental session, every participant performed two runs for each frequency in pseudo-randomized order. In the fMRI experiment, two additional runs for the 8 Hz and the 32 Hz tone (i.e., a total of four runs across frequencies) were presented in in alternating order, counterbalanced across participants. To quantify individual loudness perception, the 11 response categories were transformed into their corresponding numerical values ranging from 0 to 50 categorical units (cu) in steps of 5 cu. For each tone, the results from both runs in a session were then used to estimate individual loudness functions by means of a broken stick function with Bezier smoothing [26].

**2.3.2 Unpleasantness rating.** This task was performed in the second experimental session to assess individual unpleasantness towards the acoustic stimuli. Participants were again presented with single tone bursts and were instructed to judge each stimulus according to the question "How unpleasant did you find the sound?" via mouse click on an 11-point numeric scale ranging from 0 to 10. Two verbal labels/anchors, "Not at all" (0) and "Extremely" (10), were added to the endpoints of the scale. The question and the response scale were always visible on the screen.

The presented sound levels were calculated based on the loudness functions of each participant, corresponding to the individual loudness estimates "very soft" (5 cu), "soft" (15 cu), "medium" (25 cu), and "loud" (35 cu). If the level required to achieve any of these loudness estimates was above the upper SPL limit, the respective stimulus was excluded. All combinations of tone frequency and sound levels chosen this way were presented three times in pseudo-randomized order.

## 2.4 fMRI setup and data acquisition

The fMRI measurements were done on a 3-T scanner (Magnetom Prisma 3T, Siemens AG, Erlangen, Germany), equipped with a 20-channel head coil. The response scale and a fixation cross (see below) were projected onto a screen in the scanner bore and could be seen by the participants via a mirror construction mounted onto the head coil. To attenuate acoustic background noise produced by the MRI system, the participants' left (non-stimulated) ear was occluded with a foam earplug. Additional attenuation was provided by means of fMRI-compatible headphones (OptoACTIVE, Optoacoustics Ltd, Or Jehuda, Israel), which were also used to communicate with the participant in between measurements. Behavioral responses of

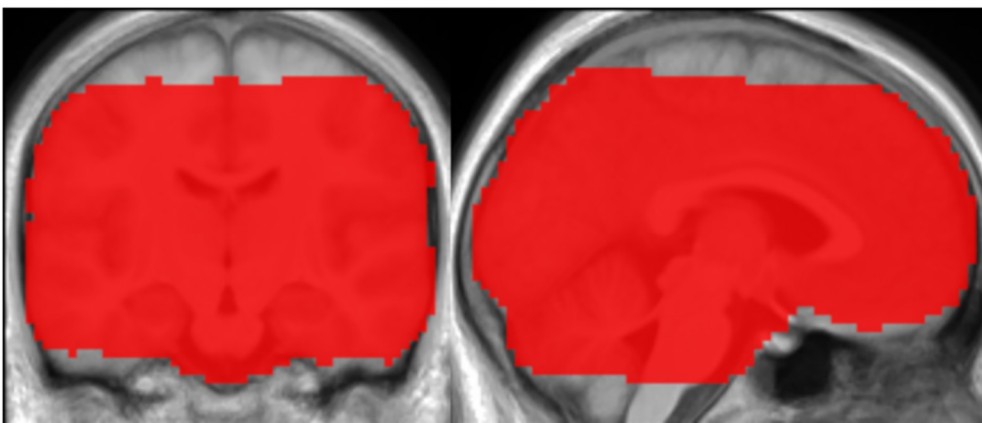

**Fig 1. Inclusive functional mask.** The map displayed in red and overlaid onto the group mean structural image covers areas included in all normalized functional images across participants. Only the voxels within this mask were analyzed in the present study.

the participants were collected via an fMRI-compatible response pad (LXPAD-2x5-10M, NAtA technologies, Coquitlam, Canada).

Functional images were obtained using a $T2^*$-weighted gradient echo planar imaging (EPI) sequence (TR 8 s, echo time 30 ms, flip angle 90°, volume acquisition time 1.5 s) with a sparse sampling paradigm to further reduce the influence of acoustic scanner noise [27, 28]. Every image comprised 26 slices (in-plane field of view 204 x 204 mm, 68 x 68 voxels, slice thickness 3 mm, gap 0.6 mm) in axial orientation, acquired in ascending interleaved order. For most participants, this volume did not fully cover the whole brain, leaving out the topmost parts of the superior frontal and parietal lobes as well as the inferior edge of the cerebellum. A mask showing the areas included in all functional images across participants is shown in Fig 1.

After completion of the functional MRI experiment, high-resolution structural images were acquired using a T1-weighted magnetization-prepared rapid acquisition gradient echo sequence (voxel size .7 x .7 x .9 $mm^3$, distance factor 50%, TR = 2 s, TE = 2.41 ms, FA = 9 °, FoV = 230 x 194 x 187 $mm^3$).

## 2.5 fMRI paradigm

The fMRI experiment was divided into two functional runs. In the first, participants completed the adaptive categorical loudness scaling task described above while auditory fMRI was performed. The task was modified so that tones were always presented four seconds after the onset of an image acquisition, with a jitter ranging from -500 to +500 milliseconds (see Fig 2, panel A). When participants did not complete the judgment of the last played tone before the onset of a scan, a silent trial followed and the presentation of the next tone was postponed to the subsequent interscan interval. Scanning was continued until a few scans after the last loudness judgment. As a consequence, the number of collected scans varied across participants. In the second functional run, participants were resented with acoustic stimuli while they performed a simple visual task, which was meant to maintain the participants' attention and monitor their wakefulness (see Fig 2, panel B). They were instructed to fixate their gaze on a small grey cross in the middle of the screen throughout the task and report when the cross changed colors from grey to either green or red, to which they responded by pressing the corresponding button on the handheld pad. These color changes always occurred around four seconds after the onset of each scan, with a jitter ranging from -500 to +500 milliseconds, and remained for a duration of

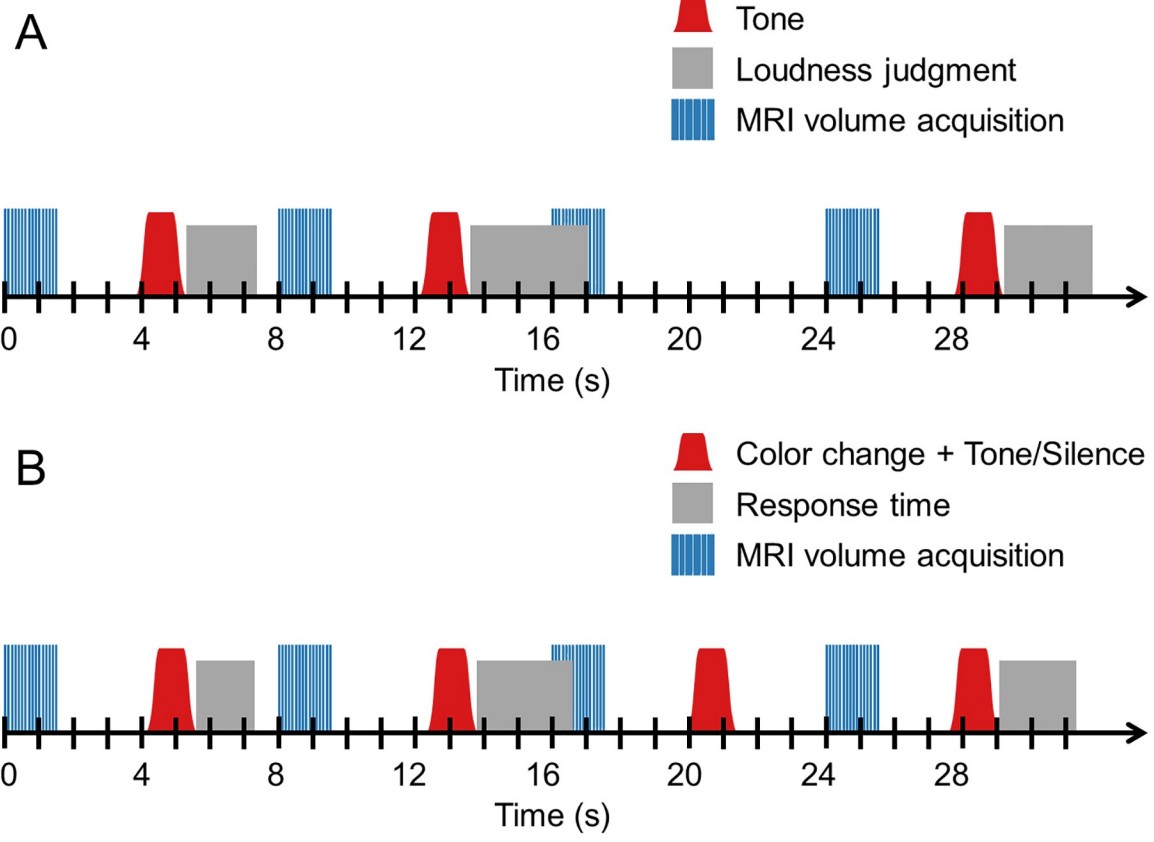

**Fig 2. Schematic depiction of the auditory fMRI paradigm.** The figure illustrates the series of events occurring across four exemplary trials in the first functional run (panel A) and the second functional run (panel B) of the fMRI experiment. The stimuli and tasks differed between both runs: In the first run, participants completed a categorical loudness scaling procedure, whereas in the second functional run, they performed a visual task while passively listening to tones (as described in the text). Apart from this, the course of events was largely similar in both runs: MR image volumes were acquired every eight seconds. Stimuli were always presented four seconds after the onset of a scan, with a jitter ranging from -500 to +500 milliseconds. Volume acquisition times and stimulus durations were 1.5 seconds. Following each stimulus presentation, participants judged the loudness of the last played tone (in the first run) or selected one of two response alternatives (in the second run) by means of button presses. In the first run, silent trials were presented when participants did not complete the judgment before the onset of the next scan. In the second run, silent trials were interspersed independently of participants' responses.

1.5 seconds. While the color was changed, one of five possible stimulus conditions was presented in pseudo-randomized order: 1) A low intensity 8 Hz tone, 2) a high intensity 8 Hz tone, 3) a low intensity 32 Hz tone, 4) a high intensity 32 Hz tone, or 5) Silence (baseline condition). Each condition was presented 40 times throughout the complete experiment. The presented levels for the low and high intensity tones were adjusted individually and aimed at corresponding to the individual loudness perceptions "very soft" (5 cu) and "loud" (35 cu), respectively. The levels required for this were derived from the loudness functions fitted to the participant's loudness judgments in the first functional run. When the level required to achieve a loudness estimate of "loud" was above the upper SPL limit, the respective tone was presented at the maximum level instead. Participants were instructed to only focus on the visual task.

### 2.6 fMRI data analysis

Preprocessing and statistical analysis of the functional imaging data was done using SPM8 (FIL, Wellcome Trust Centre for Neuroimaging, University College London, London, UK, http://www.fil.ion.ucl.ac.uk/spm) and custom scripts in MATLAB.

**2.6.1 Preprocessing.** After exclusion of the first volume, functional images were realigned to the first image of the first functional run, co-registered to the participant's structural image, normalized to Montreal Neurological Institute (MNI) space (based on information from structural tissue segmentation) and spatially smoothed with an isotropic Gaussian kernel of 8 mm full-width at half maximum. Low frequency drifts were removed by means of a high-pass filter with 1/128 Hz cut-off frequency. Possible effects of head movement and other non-stimulus related signal fluctuations were attenuated by means of nuisance regression in the process of the statistical analyses described below. For this purpose, the six estimated realignment parameters (rigid body translations and rotations), the averaged signal from voxels located in the cerebrospinal fluid (as defined by an eroded mask obtained from individual probability maps), and a binary regressor for trials with exceptionally large frame-wise signal variations ('spikes') were included as nuisance variables. For this purpose, we calculated DVARS (D referring to the temporal derivative of time courses, VARS referring to RMS variance over voxels; as described in [29]), indexing the rate of change of the BOLD signal across the entire brain at each frame of data. Spikes were defined for each participant by DVARS values more than three standard deviations above the participant's mean within each sequence.

**2.6.2 Estimates of individual loudness and unpleasantness during fMRI experiment.** For each participant, loudness estimates for every presented sound level were extracted from the individual loudness functions fitted to the loudness scaling data of the first functional run. Unpleasantness estimates for each tone frequency were obtained as follows: First, the three unpleasantness ratings for every stimulus (corresponding to a specific loudness estimate) presented in the rating task (in the booth) were averaged. Then, unpleasantness-to-loudness functions over all categorical loudness units (0 to 50) were calculated by means of simple linear inter- and extrapolation between adjacent averaged ratings, whereas all extrapolated values below 0 were set to 0 and those above 10 were set to 10 to conform to the range of the rating scale. Lastly, the unpleasantness estimates for the loudness estimates presented in the fMRI sequence were extracted from these unpleasantness-to-loudness functions.

**2.6.3 Statistical analysis.** *First functional run.* We assessed and compared the ability of sound levels, individual loudness and unpleasantness estimates to explain fMRI activation *within* each participant by means of cross-validation analyses as described in the following:

Each participant's data was split into two parts—one part contained all trials in which the 8 Hz tone was played, the other half contained all trials for the 32 Hz tone. Silent trials were excluded from this analysis. Then, separately for each data part, the data was split again into 5 parts with equal number of samples (the first 20% of samples, the following 20%, and so on) and a 5-fold cross-validation was performed: In each fold, a linear model was fitted to four parts of the data (the 'training dataset'). The resulting beta estimates were multiplied with the regressors of the remaining part (the 'test dataset') to predict the corresponding fMRI data. Collapsing over all folds, a cross-validated predicted $R^2$ was calculated. This procedure was done for three separate models that were similar to the one described above (comprised of a constant term plus linear parametric regressor), but differed with respect to the content of the parametric regressors: 1) presented levels, 2) individual loudness estimates, 3) individual unpleasantness estimates. Additionally, the procedure was also performed on a baseline model with only one constant term. The differences between models in terms of prediction performance were then statistically assessed in every voxel via one-sided paired t-tests across the predicted $R^2$ of all participants. In doing so, we investigated whether sound levels, loudness estimates, or unpleasantness estimates consistently provided better within-subject predictions across participants.

Lastly, we assessed how well the relationship between fMRI activation and level, loudness and unpleasantness for one tone was able to predict the respective relationship for the other tone:

As before, each participant's data was again split into two parts—one part containing all 8 Hz and the other half 32 Hz trials. The four models described above were fitted to each part and the resulting beta estimates were used to predict the fMRI data of the other part. Again, for every model and voxel, a predicted $R^2$ was calculated across both parts and the prediction performance of different models was compared via paired t-tests across all participants.

*Second functional run.* At the individual level, a general linear model with one binary regressor for each stimulus condition (8 Hz low intensity, 8 Hz high intensity, 32 Hz low intensity, 32 Hz high intensity), modelled as simple boxcar functions, was fitted to the time courses of every voxel. Silent trials were implicitly modelled as baseline in a constant term. To assess differences between the average responses to both tones, the respective regressors were contrasted against each other at the individual level (8 Hz– 32 Hz, and 32 Hz– 8 Hz, high and low intensity regressors were included in both contrasts). The resulting contrast maps and the first level beta maps for all conditions were then entered into one-sample t-tests at the second level.

In order to detect correlates of interindividual differences in presented level, perceived loudness, or unpleasantness, the beta maps were again entered into three separate one-sample t-tests for each stimulus condition, with 1) presented levels, 2) individual loudness estimates, and 3) individual unpleasantness estimates as covariate.

All statistical maps were thresholded either at a significance level of $p < 0.05$, corrected for family-wise-errors (FWE), or at an uncorrected significance level of $p < 0.001$, which was extended to a minimum cluster-size of at least 10 adjacent voxels. For the purpose of anatomical localization, thresholded maps were overlaid onto the group averaged structural image, using MRIcron (Version 1 2015; Chris Rorden, https://people.cas.sc.edu/rorden/mricron/).

## 3. Results

### 3.1 Loudness judgments

Individual and group averaged loudness functions for the 8 Hz and 32 Hz tone, as fitted to the participants' loudness judgments obtained in the sound booth, are shown in Fig 3. The averaged loudness functions display a nearly linear growth of loudness with sound level. Estimated detection thresholds (the lowest levels where cu > 0) were, on average, roughly 30 dB higher for the 8 Hz tone (104 dB SPL) as compared to the 32 Hz tone (72 dB SPL). Furthermore, loudness growth is noticeably steeper with sound level for 8 Hz. The individual functions reveal considerable interindividual differences in perception with respect to detection thresholds, uncomfortable loudness levels and the shape of loudness functions. With respect to the infrasonic 8 Hz tone, for instance, detection thresholds vary between 90 and almost 120 dB SPL. Likewise, the highest presented level at 140 dB elicits an uncomfortably loud acoustic event for some, yet only a medium loud percept for other participants. On average, loudness functions fitted to the judgments obtained in the MRI experiment were almost linearly related to those in the sound booth, except for a small offset in level (see S1 Fig).

### 3.2 Unpleasantness ratings

Fig 4 shows the group averaged and individual unpleasantness ratings for every participant as a function of loudness, together with the calculated inter- and extrapolations across all categorical loudness units. On average, unpleasantness grows almost linearly with loudness, and there is virtually no difference between 8 Hz and 32 Hz. By contrast, the individual participants' data vary considerably with respect to the shape of unpleasantness functions in general and

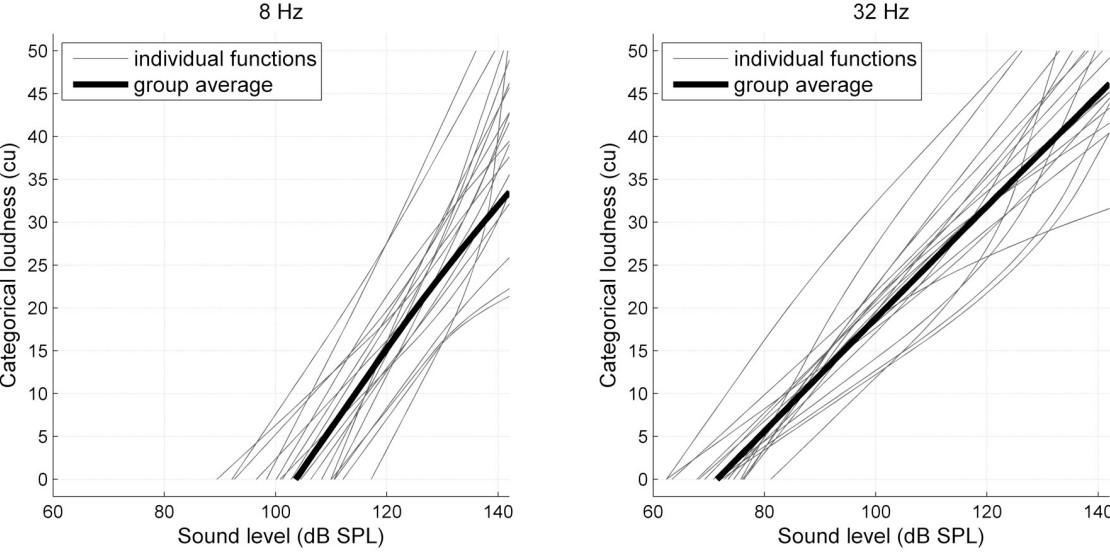

**Fig 3. Categorical loudness as a function of sound pressure level in the sound booth.** Results are shown for the 8 Hz tone (left) and the 32 Hz tone (right). The light grey curves represent individual loudness functions, the black line represents the average across all individual functions.

regarding differences between both tones. With few exceptions in the soft to medium loudness range for 32 Hz, however, unpleasantness grows monotonically with loudness over the measured dynamic range for both frequencies and across all participants.

### 3.3 General fMRI activation in response to 8 Hz and 32 Hz tones

Fig 5 shows significant responses to the four stimulus conditions in the second functional run. At low stimulus intensities, no voxels survived the family-wise error correction. At more liberal thresholds, significant clusters can be detected in left and right superior temporal lobes (STL), yet limited to lateral parts of superior temporal gyrus, including parts of the Planum temporale (PT). The extent of activation in these areas is also larger for the low intensity 32 Hz tone as compared to 8 Hz, especially in the left hemisphere. Additional clusters showing increased activation are found in the left anterior prefrontal cortex (BA 10) and left posterior cingulum/corpus callosum for 8 Hz, and the left medial precuneus for 32 Hz. At high intensities, significant clusters cover large areas of left and right STL at FWE-corrected thresholds. For both tones, these include bilaterally the posterior medial part of Heschl's Gyrus HG as well as superior temporal gyrus (anterior PT). The total extent of activation is slightly more left lateralized for the 8 Hz tone (28% more voxels in the left STL), and virtually symmetrical for 32 Hz (less than 1% difference). Despite the minor deviations in activation patterns described above, second-level t-tests revealed no significant differences between the BOLD responses to both frequencies.

### 3.4 Activation in relation to sound level and perceptual estimates

The second-level covariate analyses of the data in the second run, probing whether interindividual variation in neural responses to the stimulus conditions can be explained by means of differences with respect to presented levels, individual loudness or unpleasantness, did not yield any significant voxels at corrected thresholds. At uncorrected thresholds, significant clusters, if present, were mostly very small (below 20 voxels) and located in regions outside

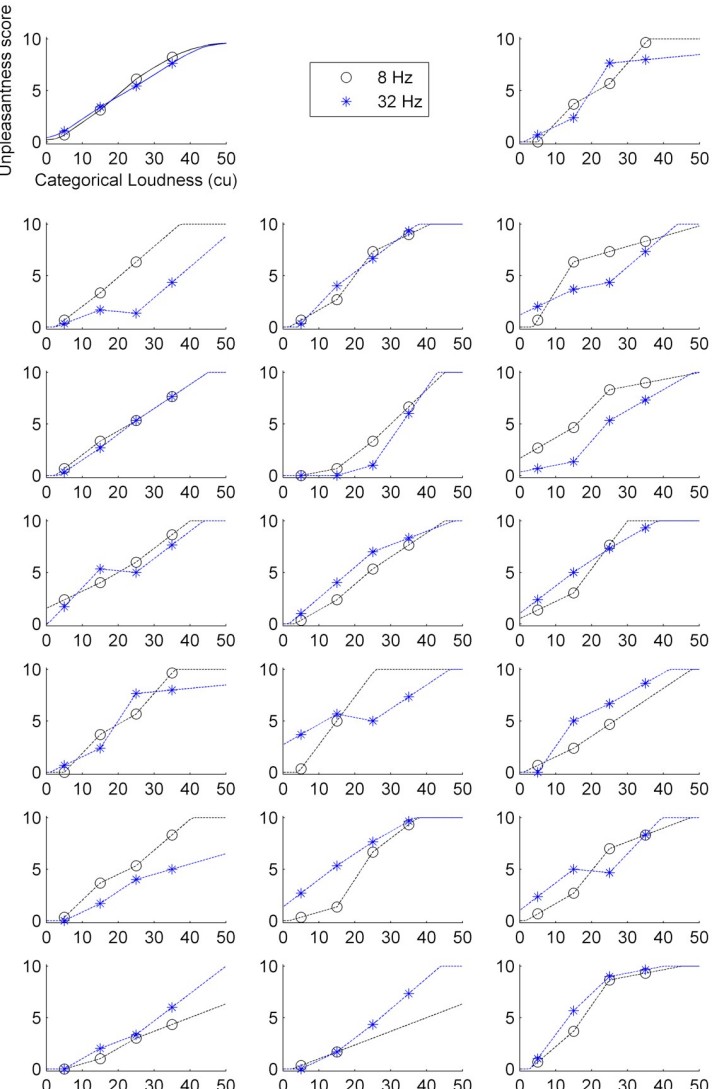

**Fig 4. Unpleasantness ratings as a function of categorical loudness.** The plot in the top left shows group averaged curves, all other plots show data of individual participants. Averaged unpleasantness ratings for every presented stimulus corresponding to a specific loudness estimate (5, 15, 25, and 35 cu, if applicable) are represented by black circles (8 Hz) and blue stars (32 Hz). Inter- and extrapolated values for all categorical loudness units are shown as black and blue lines for the 8 Hz and 32 Hztone, respectively (solid lines for the group average, dotted lines for individual data).

common auditory areas. An exception to this was only found with respect to the high intensity 32 Hz stimulus. Here, a negative correlation between activation and individual loudness was found within two moderately sized clusters in the left and right Heschl's Sulcus (48 and 72 voxels, respectively). However, since only two participants differed from the targeted perceived loudness of 35 categorical units for this stimulus—with calculated estimates of 30 cu and 36 cu (the latter being caused by the use of full dB values for the level and rounding)—this result can only be considered as anecdotal evidence, at best. In contrast, for the high intensity 8 Hz stimulus, where loudness estimates varied quite considerably across participants—from 18 cu to 36 cu—only a very small cluster (13 voxels) in the right anterior cingulum passed the uncorrected threshold.

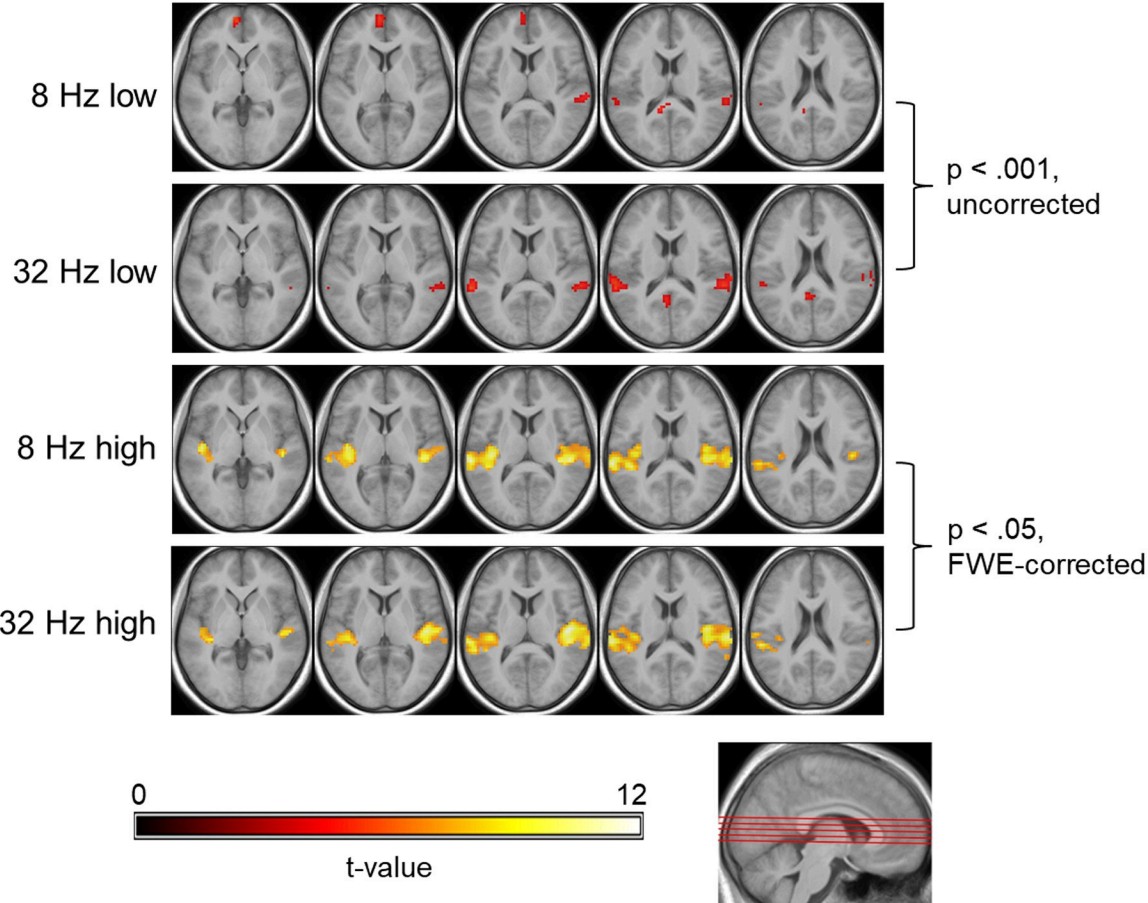

**Fig 5. Group activation in response to the four stimulus conditions in the second functional run.** From top to bottom row: 8 Hz low intensity, 32 Hz low intensity, 8 Hz high intensity, 32 Hz high intensity. For the low intensity stimuli, the second-level t-statistic maps are thresholded at a significance level of $p < 0.001$, uncorrected ($t > 3.6$), with minimum cluster-size of 10+ voxels. For the high intensity stimuli, maps are thresholded at $p < .05$, FWE-corrected ($t > 6.7$). In both cases, t-values are color-coded as indicated by the colorbar. The five axial slices are located at (from left to right) z = 0, 5, 10, 15 and 20 mm in MNI space, as illustrated by the red lines on the sagittal slice below.

The results of the within-participant cross-validation analyses are presented in Fig 6 (within-tone prediction) and Fig 7 (cross-tone prediction). Regarding the 8 Hz tone, no voxels were characterized by significantly better predictions based on presented sound levels compared to baseline. Models based on individual loudness and unpleasantness estimates however outperformed baseline models along bilateral HG and closely surrounding regions, with slightly larger clusters in the left hemisphere. Comparisons of the 'full models' (level, loudness, unpleasantness) revealed that loudness and unpleasantness estimates were also significantly superior to sound levels as a predictor of the BOLD signal in smaller parts of the aforementioned clusters (for unpleasantness, only within the left hemisphere), with a more pronounced advantage for the model with respect to loudness. No differences were found between loudness and unpleasantness. Regarding the 32 Hz tone, all three full models outperformed the baseline model in large areas of bilateral STL, including anterolateral HG and PT. Differences between the models with respect to level, loudness and unpleasantness were however similar to those for the 8 Hz tone. Specifically, models with respect to loudness and unpleasantness outperformed those with respect to level in bilateral AC. Again, there were no significant differences

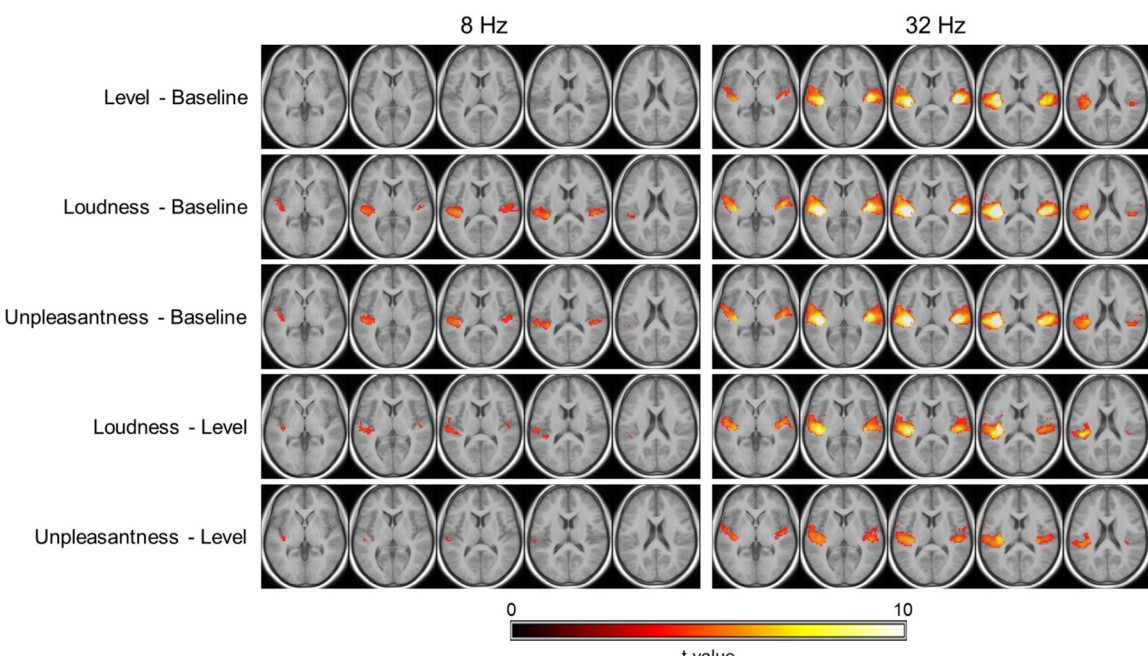

**Fig 6. Within-tone prediction performance.** Second-level t-statistic maps of the differences in cross-validated predicted R-squared between models are thresholded at p < 0.001, uncorrected (t > 3.6), with minimum cluster-size of 10+ voxels and overlaid onto the group mean structural image. The maps are color-coded by t-values as indicated by the colorbar. The five axial slices are located at the same z-coordinates as in Fig 5.

between loudness and unpleasantness. Noticeably higher t-values in the loudness versus level contrast compared to the unpleasantness versus level contrast still indicate a slight advantage for the models with respect to loudness.

As for the cross-tone prediction, significant clusters for all three full models (compared to baseline) were largely similar to those of the within-tone analysis for 32 Hz. This did not hold for the most anterolateral parts of HG in both hemispheres, where prediction accuracy was not higher for the full models than for the baseline model. The comparison of the full models yielded similar results as in the within-tone analysis with respect to the 8 Hz tone: Loudness outperformed level in the majority of voxels along the left HG and within a small cluster in the right hemisphere, while unpleasantness was superior to level only in a very small number of voxels around left posteromedial HG. Lastly, the analysis revealed a significant advantage of loudness over unpleasantness, yet limited to a tiny cluster above the central part of left HG.

## 4. Discussion

The results of this study can be summarized as follows: (1) Normal hearing listeners were able to detect LFS (32 Hz) and IS (8 Hz) tone bursts in a sound booth as well as in the MRI scanner and could evaluate them in terms of perceived loudness, with similar results in both environments. Specifically, detection thresholds were higher and loudness growth with sound level was steeper for 8 Hz as compared to 32 Hz. Unpleasantness ratings in the booth increased almost linearly as a function of loudness. These functions were similar for both tones when averaged across participants, but revealed considerable interindividual differences. (2) Significant fMRI activation in response to both tones was mainly found in bilateral primary and secondary AC. Activation patterns were not significantly different between 32 Hz and 8 Hz. (3) Between-participants variation in fMRI activation was not correlated with interindividual

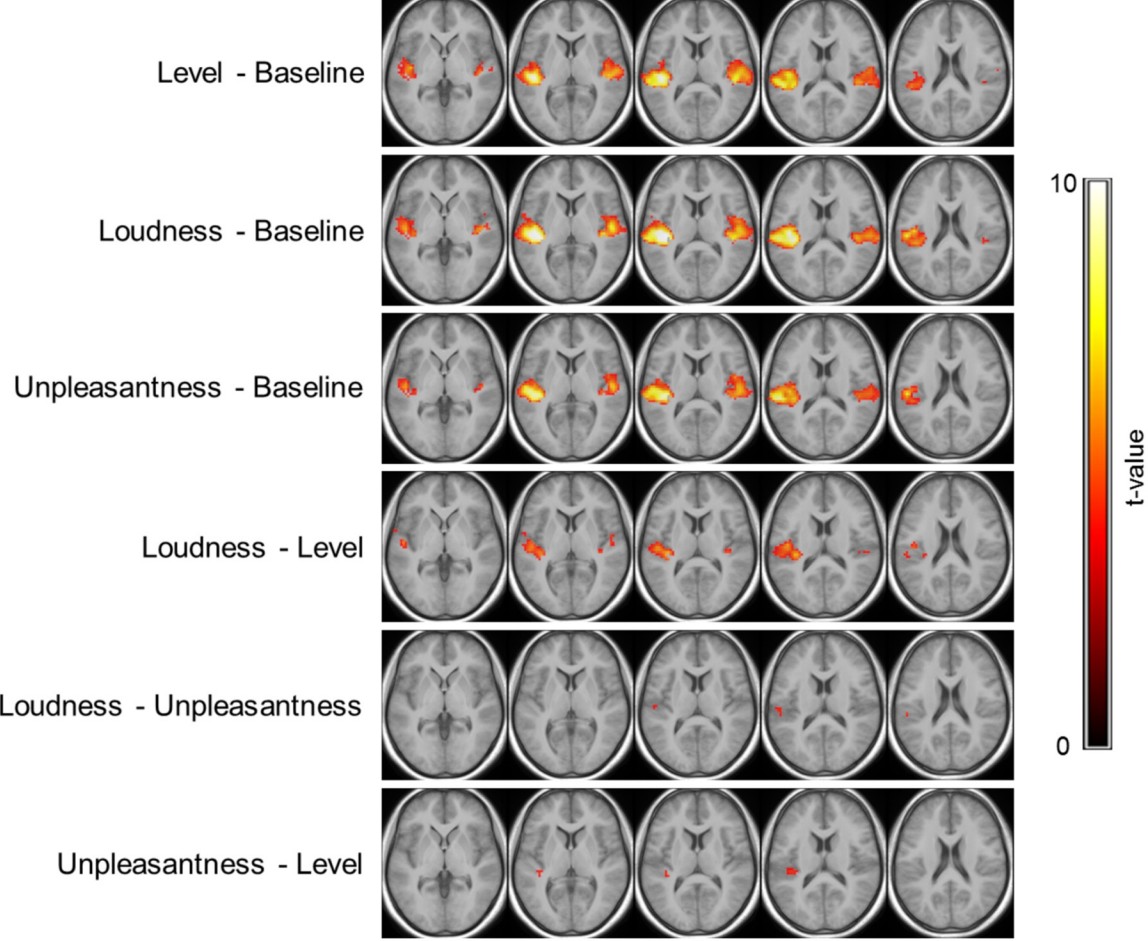

**Fig 7. Cross-tone prediction performance.** Second-level t-statistic maps of the differences in cross-validated predicted R-squared between models are thresholded at p < 0.001, uncorrected (t > 3.6), minimum cluster-size of 10+ voxels and overlaid onto the group mean structural image. The maps are color-coded by t-values as indicated by the colorbar. The five axial slices are located at the same z-coordinates as in Fig 5.

differences with respect to loudness or unpleasantness. (4) Within-participants variation in bilateral AC activation was successfully predicted by sound levels and estimates of individual loudness and unpleasantness based on information from other trials of the same tone and even based on data from the other tone only. In both cases, loudness outperformed unpleasantness and level in terms of prediction accuracy.

## 4.1 Behavioral results

Our behavioral results are largely in line with expectations derived from the literature. First of all, estimated detection thresholds (slightly above 70 and 100 dB SPL for 32 Hz and 8 Hz, respectively) were comparable to those found in previous investigations [2–5]. Secondly, our data are in line with several other studies suggesting that the growth of perceived loudness with level steepens as frequency decreases from very low into infrasonic frequencies [11–13]. As such, the general trend seen for frequencies below 1 kHz [30] continues, leading to a very narrow perceptual dynamic range, i.e., the level range from just audible to uncomfortably loud, for infrasound. Likewise, unpleasantness grew more steeply with level for the lower

frequency tone, conforming to previously published data [4, 14]. The latter finding is not easily discernable from Fig 4, which only shows unpleasantness as a function of loudness, with virtually no difference between the slopes for both tones. It can however be inferred by considering that loudness grew much steeper as a function of level for 8 Hz than for 32 Hz.

Loudness functions for the LFS and IS tones used in this study were much more linear than those typically found for tones at higher frequencies. This is in line with the findings reported by Kühler et al. [13]. For narrowband signals at higher frequencies, loudness functions are typically characterized by a steepening of slope with increasing level. This nonlinearity is attributed to active compression or gain mediated via outer hair cells (OHC) in the cochlea [31]. Conversely, a more linear loudness function as often observed in hearing impaired listeners is indicative of a loss of OHC function [32]. Following this thought, there is apparently no effect of OHC activity at very low and infrasonic frequencies. The physiological mechanisms of IS processing in the cochlea are not yet fully understood. There is however strong evidence that IS at high levels enters the inner ear and leads to a modulation of the spontaneous firing rate of auditory nerve cells [33] as well as periodic changes with respect to the operating point of the cochlear amplifier [8]. Ultimately, this can alter the processing and perception of sound at higher frequencies and under certain conditions mimic an amplitude modulation of other audible sounds [34].

What's more important regarding the aim of the present study, our data revealed considerable interindividual differences between participants. Estimated detection thresholds varied over a range of 20 dB for the 32 Hz tone and close to 30 dB for the 8 Hz tone. Beyond that, participants differed with respect to the shapes of loudness functions, with increasing, flat and declining slopes. Similarly, there were noticeable differences with respect to the shapes of unpleasantness functions. This conforms to observations made in studies of low frequency noise suggesting pronounced variation in the extent of annoyance [16, 17] and its relation to sound level [35]. In another study, Broner and Leventhal [36] compared magnitude estimations of loudness and annoyance given by 20 listeners (10 males and 10 females) for noises with 10 Hz bandwidths between 20–90 Hz. At group level, they found no difference between the growth of loudness and the growth of annoyance with level. However, there were significant differences between male and female participants. Moreover, they reported substantial variation in the loudness-to-annoyance magnitude ratios across listeners (which the authors corrected for before performing their analyses). Despite the lack of analyses regarding sex differences here, our data share a common trend with those of Broner and Leventhal [36]. Specifically, unpleasantness grew nearly linear with loudness for both tones in terms of the group average, but deviated markedly at individual level. Naturally, part of the interindividual differences might be attributable to different strategies employed by the participants to evaluate the sounds. Plus, there is certainly a higher level of uncertainty for the unpleasantness estimates as compared to the categorical loudness estimates due to the lower number of presented stimuli, especially in those listeners that were limited to three or even only two different levels of the 8 Hz tone. Notwithstanding these concerns, we argue based on our data that individual unpleasantness for LFS and IS cannot be directly inferred from the perceived loudness of the sound and that the two respective estimates to some degree reflect different qualities of sound perception.

Although a small number of studies have presented LFS and IS while performing fMRI before [9, 10, 37], to our knowledge this is the first study to assess perception of the participants for these sounds during the scanning procedure. Our data indicate that loudness perception is comparable in both environments (see S1 Fig), at least when presenting stimuli during silent intervals in a sparse sampling paradigm. This conforms to Röhl and Uppenkamp [38], who measured loudness functions for a broadband noise stimulus in the sound booth and the

MRI scanner and found no significant differences between them. Whether the same is true with regard to the perceived unpleasantness of the LFS and IS stimuli could not be confirmed in the present study. Unlike loudness judgments, individual unpleasantness ratings were only assessed in the sound booth.

## 4.2 General fMRI activation

To our knowledge, only two other auditory fMRI studies have reported changes of activation in response to LFS and IS at the time of writing of this manuscript [9, 10], with another study focusing solely on changes of connectivity between brain regions [37]. Dommes et al. [9] found significant activation in primary and secondary AC (Brodmann areas 41, 42 and 22) in response to short tone bursts at 12 Hz with sound pressure levels of 120 and 110 dB and at 48 Hz with 100 dB, which resembled activation patterns for a 500 Hz tone with 105 dB that they included as a control. Weichenberger et al. [10] also used a 12 Hz tone, with individually adjusted levels to achieve a "medium" loudness sensation, which resulted in an average applied sound pressure level of 115 dB. Similar to Dommes et al. [9], their data revealed increased activation exclusively in primary AC (BA 41 and 42).

In the present study, there were only small clusters of activation (mainly in PT) for 8 Hz and 32 Hz tones at "low intensity", i.e. when sound pressure levels were just above threshold (83 dB for 32 Hz and 111 dB for 8 Hz, averaged across participants). In contrast, large areas of superior temporal lobes including primary and secondary AC were activated at "high intensity" (129 dB for 32 Hz and 138 dB for 8 Hz). In this regard, our results agree well with the aforementioned studies and support the idea that, at cortical level, infrasound is processed similar to sounds in the typical audio frequency range. The lack of significant differences between responses to the LFS and the IS tone underscores that the physiological mechanisms in the human brain forming the basis of perception do not suddenly change in a substantial way when the sound frequency is lowered into the infrasonic range.

By contrast, our results differed considerably from those of the two previous studies in terms of lateralization of activation. Dommes et al. [9] and Weichenberger et al. [10] presented their stimuli monaurally to the right ear of their participants, just as in the present study. Based on fMRI activation for sounds in the typical audio frequency range [e.g. 39, 40], one would expect much stronger contralateral as compared to ipsilateral activation at cortical level. This is exactly what Dommes et al. [9] and Weichenberger et al. [10] reported. In fact, Dommes et al. [9] detected about three times as many activated voxels in the contralateral AC for the 48 Hz tone and even four and eight times as many at 12 Hz with 110 and 120 dB, respectively. In the present study, activation was much less lateralized for the 8 Hz stimulus and virtually symmetrical for 32 Hz. Notably, we obtained similar results in this regard across two functional runs and with different paradigms and analyses. The origin of the differences between the present and previous findings remains elusive at this point.

## 4.3 Activation in relation to sound level, loudness and unpleasantness

A large body of evidence supports that fMRI activation in AC increases as a function of the sound level or loudness of sounds in the typical audio frequency range. This has been demonstrated for pure tones [41, 42], frequency modulated tones [43, 44], and three-tone patterns [45], narrowband noise [46] and broadband noise [38, 47, 48], as well as speech stimuli [49]. The present results extend the existing literature and suggest that the same is true for tones in very low and infrasonic frequencies. Specifically, fMRI activation in primary and secondary AC was significantly predicted by regression models that assumed a linear relationship between the activation magnitude and level (for 32 Hz) or loudness estimates (for both tones).

Similar to previous studies that presented stimuli monaurally [e.g. 42, 46], as in the present study, this relationship was significant in the contralateral as well as the ipsilateral AC.

The cross-validation analyses also revealed that activation in the AC of individual participants was more closely related to the respective subject's perception of tones as compared to presented levels. Individual loudness and unpleasantness estimates predicted the individual fMRI signal in AC significantly better than level, based on "learned" information from other trials during the experiment. This was the case when the "training data" was taken from trials of the same tone (as in the predicted data), but also when it only contained data from trials of the tone at the other frequency. Under both conditions, individual loudness estimates performed best out of the three variables (level, loudness, unpleasantness) in terms of prediction accuracy, yet only slightly (and mostly non-significantly) better than estimates of unpleasantness. Notably, as mentioned above, unpleasantness was calculated from data obtained in the sound booth, whereas loudness was also assessed in the MRI environment. Hence, unpleasantness estimates were subject to a comparatively higher level of uncertainty.

These findings agree with several reports of other auditory fMRI studies, which suggest that activation in AC is more closely related to perceived loudness rather than level [e.g. 38, 44, 46] or other physical sound parameters such as the bandwidth [50]. Here again, the present results suggest that insights gained with respect to fMRI activation in response to sounds in the typical audio frequency range also pertain to IS.

Interestingly, voxels in which activation could be explained significantly better than baseline by any of the three variables were (virtually) exclusively found in the STL. Moreover, there were no regions with higher prediction accuracy for unpleasantness as compared to loudness. As mentioned in the introduction, several fMRI studies suggested that the amygdala might play a prominent role in the processing of unpleasantness [20, 51]. Accordingly, one might have expected that unpleasantness estimates should provide significant predictions of activation in this region. It should be noted, however, that studies speaking towards a link between perceived unpleasantness and the amygdala (including the aforementioned) usually compared activation in response to different environmental sounds or orchestra music. Consequently, stimuli not only differed considerably with regard to their spectrotemporal content, but, more importantly, also in terms of their emotional valence. This is in stark contrast to the stimuli in the present study, which only varied in terms of their intensity and likely had little emotional associations attached to them, such that the participants could essentially only judge them by their *sensory* unpleasantness. It is of course theoretically possible for a listener to develop a learned aversion for previously neutral sounds (as a result of past events), which might ultimately lead to elevated unpleasantness when exposed to these sounds. Findings from research in mice [21] and human listeners [22] suggest that such a learned aversive valence for acoustic signals may then be reflected by neural responses in the AC. Given that activation in AC was not better explained by unpleasantness relative to the other two variables in the present study, though, our data do not support the idea that learned aversion played an important role with respect to listeners' unpleasantness ratings.

Despite the clear relationship between perceptual measures and fMRI activation at the individual level, interindividual differences with respect to loudness or unpleasantness were not correlated with differences in terms of neural responses to the stimuli in the second run. The lack of significant results concerning loudness is probably not surprising when considering that the levels for individual stimulus conditions were specifically chosen to correspond to the same loudness sensation for every participant. However, while this objective was fulfilled for thee out of four stimulus conditions in the experiment, there was still considerable variation across listeners for the high intensity 8 Hz tone. Since this stimulus was rarely ever judged as "loud", it was instead presented at the upper sound pressure limit to the majority of

participants, which corresponded to estimated loudness sensations ranging from 18 cu (just above "soft") to 36 cu ("loud"). Likewise, unpleasantness estimates varied strongly across listeners for this condition (from 3 to 11 on the 11-point scale), but they also displayed moderate variation for all other conditions. Nonetheless, the present data did not reveal physiological correlates of these perceptual differences.

In another group of normal hearing listeners, Röhl and Uppenkamp [38] found a significant correlation between individual loudness estimates and activation in the AC at a fixed sound pressure level of 80 dB for a pink noise stimulus. The 80 dB stimulus was chosen amongst several other presented levels based on the fact that it was the highest level presented to all participants in that study. Consequently, BOLD signal changes were relatively large, and loudness presumably varied most widely across listeners (from about 25 to 41 cu). Hence, the conditions were strikingly similar to those for the high intensity 8 Hz stimulus in the present study. Naturally, there were still small variations of level for the high intensity 8 Hz tone (from 132 to 140 dB), but these were much less pronounced than for the other three conditions, where levels varied about 30 dB across participants.

The discrepancy between the present results and those of Röhl and Uppenkamp [38] therefore raises the possibility that the representation of individual loudness perception in the auditory system for IS tones may after all be different from broadband noise in the typical audio frequency spectrum. One caveat to this conclusion is that the number of participants in the present study (n = 19) was considerably smaller than that of Röhl and Uppenkamp [38] (n = 45). It is possible that a larger sample could unveil a significant relationship between individual differences in perception and brain activation for LFS and IS tones, which, in this study, was swamped by other unmodeled factors causing interindividual variation of the fMRI signal (e.g. neurovascular coupling).

Further measures to improve detection power in future studies include specific care concerning the selection of the presented stimuli, so that variation in the dimension of interest (e.g. loudness or unpleasantness) is maximized. This however poses a particular challenge with respect to IS, considering the pronounced differences across listeners in terms of hearing thresholds and the markedly limited perceptual dynamic range. Another way to increase the variation would be to include individuals that are (self-reportedly) characterized by an abnormal sensitivity to LFS and IS, such as in the study of Inukai et al. [18].

Lastly, it should be noted that the conditions in this study differ in many respects from the typical environmental setting in which people are exposed to LFS and IS. In the latter, the affected person is usually exposed over longer periods of time, sounds are more complex than simple tones, and they target the whole body as opposed to only one ear. The present study specifically addressed the processing of LFS and IS in the auditory system. Hence, stimuli were delivered directly in the ear canal in order to avoid possible confounds due to other bodily reactions at high levels under free-field conditions (i.e., whole body stimulation). At this stage, the ecological validity of the present findings from our laboratory study is limited. Still, our results indicate that auditory functional MRI is a useful way to identify neural correlates of the individual perception of LFS and IS.

## Conclusions

The auditory fMRI data obtained in this study substantiate the notion that airborne LFS and IS are processed in the human central auditory system, similar to sounds in the typical audio frequency range. There was considerable interindividual variation among listeners with respect to judgments of perceived loudness and unpleasantness for the LFS and IS tones. Neural correlates of these perceptual differences could not yet be identified. Still, at the level of individual

listeners, activation in the AC appears to be more closely related to individual loudness and unpleasantness rather than sound level.

## Supporting information

**S1 Fig. Comparison of loudness functions in the MRI environment versus the sound booth.**
(PDF)

## Acknowledgments

The authors would like to thank all participants. The authors also thank Kirsten Netter (Universität Oldenburg) for performing parts of the psychoacoustical measurements, Elisa Burke and Johannes Hensel for technical support with respect to the infrasound source provided by PTB Braunschweig, as well as Christian Koch (PTB Braunschweig) and all project partners across Europe within the EARS-II project for many fruitful discussions.

## Author Contributions

**Conceptualization:** Oliver Behler, Stefan Uppenkamp.

**Data curation:** Oliver Behler.

**Formal analysis:** Oliver Behler.

**Funding acquisition:** Stefan Uppenkamp.

**Investigation:** Oliver Behler.

**Methodology:** Oliver Behler.

**Project administration:** Stefan Uppenkamp.

**Supervision:** Stefan Uppenkamp.

**Visualization:** Oliver Behler.

**Writing – original draft:** Oliver Behler.

**Writing – review & editing:** Oliver Behler, Stefan Uppenkamp.

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
