## [Decision Letter · Decision Letter 0]

4 Dec 2019

PONE-D-19-22285

Activation in human auditory cortex in relation to the loudness and unpleasantness of low-frequency and infrasound stimuli

PLOS ONE

Dear Dr. Behler

We have now received two reviews for your manuscript.  Reviewer 1 suggests a major revision, and would like to see a clearer definition of the aims and subsequent analyses to strengthen the manuscript, and Reviewer 2 suggests clarification of the interpretation of the results. 

Both reviewers find the study to be of merit and interest, and my view the reviewers' points can be addressed as a minor revision.

Therefore, we invite you to submit a revised version of the manuscript that addresses the points raised during the review process.

We would appreciate receiving your revised manuscript within 45 days of this decision date. To enhance the reproducibility of your results, we recommend that if applicable you deposit your laboratory protocols in protocols.io, where a protocol can be assigned its own identifier (DOI) such that it can be cited independently in the future. For instructions see: http://journals.plos.org/plosone/s/submission-guidelines#loc-laboratory-protocols

We look forward to receiving your revised manuscript.

Kind regards,

Ifat Yasin

Academic Editor

PLOS ONE

Journal Requirements:

Reviewers' comments:

Reviewer's Responses to Questions

**Comments to the Author**

1. Is the manuscript technically sound, and do the data support the conclusions?

Reviewer #1: Yes

Reviewer #2: Yes

2. Has the statistical analysis been performed appropriately and rigorously? 

Reviewer #1: Yes

Reviewer #2: Yes

3. Have the authors made all data underlying the findings in their manuscript fully available?

Reviewer #1: Yes

Reviewer #2: Yes

4. Is the manuscript presented in an intelligible fashion and written in standard English?

Reviewer #1: No

Reviewer #2: Yes

5. Review Comments to the Author

Reviewer #1: In the current study, Behler & Uppenkamp examined behavioral and neural responses to acoustic stimuli with varying veridical loudness, perceived loudness, and perceived pleasantness. To this end they presented healthy participants with low frequency sounds (LFS; < 200Hz) and infrasonic sounds (IS; < 20Hz) monaurally to the right ear. The stated aims of the study are:

(1) Are perceived loudness and unpleasantness distinctly represented in the human brain?

(2) Can interindividual differences with respect to loudness and unpleasantness for LFS and IS be identified in terms of objective, physiological correlates?

Throughout the paper, the stated aims get mixed (and side-tracked) with analysis focusing on alternative questions.

The authors go to a lot of effort to show that the perceptual curves outside/inside the scanner are similar (which is interesting by itself), and dedicate a lot of effort in the comparison of 8Hz and 32Hz stimuli. These are different questions from the stated framework of the study that left me confused with respect to what is the scientific question that the authors are trying to answer. Another example - what is the logic of presenting sounds monaurally? Is this part of the author’s question (bilateral vs. contralateral activations)? When the question is unclear, the logic of the experimental design is also unclear and the focus of the paper gets lost.

The presented data demonstrates:

1) Increase in absolute sound level (SPL) results in increased perceived loudness (figure 2). This is to be expected.

2) There is correspondence between perceived loudness assessed outside/inside the fMRI scanner (figure 3). This is an important and novel observation to the best of my knowledge, although subsidiary with respect to the stated aims.

3) Figure 4 suggests that IS are reported as more unpleasant than LFS although at the group level there is no difference.

4) Figures 5 & 6 & 7 demonstrate activation in auditory cortex to the 8 and 32Hz stimuli that increases with sound loudness.

5) Figures 8 & 9 pertain directly to the stated aims and suggest that perceived loudness and degree of pleasantness are more closely related to the degree of fMRI activation in auditory cortex as compared to absolute sound level.

I have no technical reservations but in my opinion a re-framing of the manuscript would help strengthen the manuscript and allow the authors to address their stated aims in a more focused manner.

General comments:

1) A schematic figure showing when stimuli were delivered with respect to fMRI sampling and subject motor responses would be helpful.

2) Titles for the figures could facilitate readability.

3) For the stated aims it seems that MEG (or some other method with a more acoustically quiet recording environment) would have been more appropriate.

Reviewer #2: In this fMRI study, Behler and Uppenkamp investigated the neural basis of the audition of low frequency sounds (LFS, 32 Hz) and infrasounds (IS, 8 Hz) in relation to subjective estimates of sensory loudness and unpleasantness.

The authors found a high between-participants variability in detection thresholds, loudness functions and unpleasantness functions. On the neural level, the authors found significant tone-evoked activation patterns in the bilateral primary and secondary auditory cortices, which did not differ between 32 Hz and 8 Hz sounds and which increased as a function of sound physical level. Moreover, they did not observe a significant correlation between the interindividual differences in loudness or unpleasantness ratings and the variation of the fMRI activation. Within-participants differences in the bilateral activation patterns of the auditory cortices were predicted by sound levels, unpleasantness ratings and, especially, loudness ratings.

The study is well-designed, technically well-implemented and the results are interesting. However, I have some concerns which the authors should take into account:

[1] The first point is referred to the fMRI experiment. During the second functional run, participants were presented with 8 Hz and 32 Hz tones, whose low/high intensities were individually adjusted according to the loudness ratings provided by each participant during the first functional run. However, authors only collected unpleasantness judgements during the behavioural experiment, which was performed outside the MRI scanner (as stated in the Methods section 2.6.2). Even if the loudness functions in both conditions were linearly related, how can authors assume that the individual unpleasantness estimates in the noisy MRI environment (not collected) would have corresponded to those collected outside the MRI scanner (i.e. inside the silent booth)?

Moreover, in the Discussion the authors stated that “Normal hearing listeners were able to detect LFS (32 Hz) and IS (8 Hz) tone bursts in a sound booth as well as in the MRI scanner and could evaluate them in terms of perceived loudness and unpleasantness, with similar results in both environments” but unpleasantness was not assessed in the MRI environment.

[2] Concerning the unpleasantness, one recent fMRI study (Staib et al., Hum Brain Mapp 2019) found a significant activation of the human primary auditory cortex for different types of sounds with a learned emotional aversiveness. Moreover, in the rodent secondary auditory cortex there have been observed neurons that selectively respond to stimuli with a learned aversive valence (Grosso et al., Nat Comm 2015). Even if not overlapping with the notion of pure sensory unpleasantness, the authors may also wish to discuss these studies.

[3] The authors may also wish to include in the Discussion the perspective reported in the work of Monson, Han and Purves (PLoS ONE 2013).

[4] One typo: “resented” instead of “presented” (line 187).

6. PLOS authors have the option to publish the peer review history of their article (what does this mean?). If published, this will include your full peer review and any attached files.

Reviewer #1: No

Reviewer #2: Yes: Benedetto Sacchetti (with the assistance of Eugenio Manassero)

---

## [Author Response · Author response to Decision Letter 0]

16 Jan 2020

We’d like to thank the reviewers for the constructive and useful feedback on the previous version of the manuscript. In our view, the changes applied on the basis of the reviewers’ comments have considerably improved the quality of the manuscript. The detailed responses to all individual comments are included in the separate file labeled "Response to Reviewers".

---

## [Decision Letter · Decision Letter 1]

30 Jan 2020

Activation in human auditory cortex in relation to the loudness and unpleasantness of low-frequency and infrasound stimuli

PONE-D-19-22285R1

Dear Dr. Behler

We are pleased to inform you that your manuscript has been judged scientifically suitable for publication and will be formally accepted for publication once it complies with all outstanding technical requirements.

With kind regards,

Ifat Yasin

Academic Editor

PLOS ONE

Additional Editor Comments (optional):

Reviewers' comments:

Reviewer's Responses to Questions

**Comments to the Author**

1. If the authors have adequately addressed your comments raised in a previous round of review and you feel that this manuscript is now acceptable for publication, you may indicate that here to bypass the “Comments to the Author” section, enter your conflict of interest statement in the “Confidential to Editor” section, and submit your "Accept" recommendation.

Reviewer #2: All comments have been addressed

2. Is the manuscript technically sound, and do the data support the conclusions?

Reviewer #2: Yes

3. Has the statistical analysis been performed appropriately and rigorously? 

Reviewer #2: Yes

4. Have the authors made all data underlying the findings in their manuscript fully available?

Reviewer #2: Yes

5. Is the manuscript presented in an intelligible fashion and written in standard English?

Reviewer #2: Yes

6. Review Comments to the Author

Reviewer #2: Behler and Uppenkamp adequately addressed all my comments. Therefore my recommendation is the acceptance of the manuscript for publication.

7. PLOS authors have the option to publish the peer review history of their article (what does this mean?). If published, this will include your full peer review and any attached files.

Reviewer #2: Yes: Benedetto Sacchetti, with the assistance of Eugenio Manassero

---

## [Editor Report · Acceptance letter]

7 Feb 2020

PONE-D-19-22285R1 

Activation in human auditory cortex in relation to the loudness and unpleasantness of low-frequency and infrasound stimuli 

Dear Dr. Behler:

I am pleased to inform you that your manuscript has been deemed suitable for publication in PLOS ONE. Congratulations! Your manuscript is now with our production department. 

With kind regards,

on behalf of

Dr. Ifat Yasin 

Academic Editor

PLOS ONE